# Determinants of Erythrocyte Lead Levels in 454 Adults in Florence, Italy

**DOI:** 10.3390/ijerph16030425

**Published:** 2019-02-01

**Authors:** Saverio Caini, Benedetta Bendinelli, Giovanna Masala, Calogero Saieva, Melania Assedi, Andrea Querci, Thomas Lundh, Soterios A. Kyrtopoulos, Domenico Palli

**Affiliations:** 1Cancer Risk Factors and Lifestyle Epidemiology Unit, Institute for Cancer Research, Prevention and Clinical Network (ISPRO), 50139 Florence, Italy; b.bendinelli@ispro.toscana.it (B.B.); g.masala@ispro.toscana.it (G.M.); c.saieva@ispro.toscana.it (C.S.); m.assedi@ispro.toscana.it (M.A.); andrea_1975_2@hotmail.it (A.Q.); d.palli@ispro.toscana.it (D.P.); 2Division of Occupational and Environmental Medicine, Lund University Hospital, 22363 Lund, Sweden; thomas.lundh@med.lu.se; 3National Hellenic Research Foundation, Institute of Biology, Pharmaceutical Chemistry and Biotechnology, 11635 Athens, Greece; skyrt@eie.gr

**Keywords:** lead, determinant, diet, lifestyle, Italy

## Abstract

*Background*: Lead exposure, even at low levels, is associated with adverse health effects in humans. We investigated the determinants of individual lead levels in a general population-based sample of adults from Florence, Italy. *Methods*: Erythrocyte lead levels were measured (using inductively coupled plasma-mass spectrometry) in 454 subjects enrolled in the Florence cohort of the European Prospective Investigation into Cancer and Nutrition (EPIC) study in 1992–1998. Multiple linear regression models were used to study the association between demographics, education and working history, lifestyle, dietary habits, anthropometry, residential history, and (among women) menstrual and reproductive history and use of exogenous sex hormones, and erythrocyte lead levels. *Results*: Median lead levels were 86.1 μg/L (inter-quartile range 65.5–111.9 μg/L). Male gender, older age, cigarette smoking and number of pack-years, alcohol intake, and residing in urban areas were positively associated with higher erythrocyte lead levels, while performing professional/managerial or administrative work or being retired was inversely associated with lead levels. Among women, lead levels were higher for those already in menopause, and lower among those who ever used hormone replacement therapy. *Conclusions*: Avoidable risk factors contribute to the lead body burden among adults, which could therefore be lowered through targeted public health measures.

## 1. Introduction

Lead (Pb) is a ubiquitous heavy metal with several unique properties (malleability, ductility, low melting point, and resistance to oxidation and corrosion) which have resulted in extensive usage in industry, for instance in construction and plumbing; for the production of paints, pigments, ammunitions, and lead-acid batteries; as an antiknock agent in leaded gasoline; and for radiation protection [1]. Lead is non-biodegradable and persists indefinitely in the environment, and humans can become exposed to it mainly via inhalation and ingestion. Once absorbed, lead is distributed throughout the body and mainly deposited in bones and teeth, and interferes with normal cell function and several physiological processes [1,2,3]. 

Exposure to lead has been associated with a wide range of adverse health effects in humans. Chronic exposure to high levels of lead (which usually occurs only among professionally exposed individuals) can produce lead nephropathy, i.e., a chronic tubulointerstitial nephritis which may cause renal failure and require dialysis or transplantation [4]. Prolonged exposure to low lead levels is also a cause for concern globally, as it can induce changes in glomerular filtration rate and may result in chronic kidney disease [4,5,6]. In addition to renal function impairment, lead toxicity effects include central nervous system and neuromuscular manifestations (e.g., loss of short-term memory and concentration, sleep disturbances, and extensor muscle weakness), gastrointestinal symptoms (e.g., abdominal colic), hepatobiliary damage, anaemia, and others [2,7]. Worryingly, the health effects of lead exposure can occur early in childhood and extend throughout an individual’s life [8,9]. Finally, inorganic lead compounds are classified as probably carcinogenic to humans (Group 2A of the International Agency for Research on Cancer (IARC) classification), with suggestive evidence of an increased risk for stomach and lung cancer among professionally exposed workers [10]. 

Because of the pervasive distribution of lead in the modern environment, the multiplicity of sources and routes of exposure, and the ability to bioaccumulate in the human body and produce negative health effects even at low exposures, lead pollution represents an important public health problem globally, and the in-depth study of the main predictors of lead body levels is an essential step before effective measures aimed to mitigate exposure to it can be implemented. Here, we aimed at investigating the main determinants of individual lead levels in a population-based series of 454 adults from Tuscany, central Italy. 

## 2. Materials and Methods

### 2.1. Study Population and Data Collection

The present study was carried out within the Florence arm of the European Prospective Investigation into Cancer and Nutrition (EPIC), an ongoing multicentre cohort study whose main objective is to explore the role of diet, lifestyle and environment in the aetiology of cancer [11,12]. A total of 13,597 cancer-free volunteers aged 35–65 years (74.2% women), mostly residing in the Provinces of Florence and Prato, were enrolled into the Florence-EPIC study between 1992 and 1998 [13]. 

The present study is based on members of the EPIC-Florence cohort that were later included in Envirogenomarkers (EGM), a nested case-control study carried out in 2009 with the purpose of investigating the role of common environmental pollutants (including lead) in the aetiology of breast cancer and non-Hodgkin lymphoma [14]. A total of 197 breast cancer cases and 31 non-Hodgkin lymphoma cases from the EPIC-Florence cohort were included in the EGM study, along with 228 controls individually matched to cases on a 1:1 ratio based on gender, age at recruitment, and date of blood collection. 

### 2.2. Data Collection

All EPIC Florence participants filled a locally-tailored food frequency questionnaire that investigated dietary habits over the last 12 months, and a lifestyle questionnaire that contained questions on smoking habits, intake of alcoholic beverages, education and socioeconomic status, past medical history, physical activity levels, and (for women) details of menstrual and reproductive history and use of oral contraceptives and hormone replacement therapy. Study participants also had their anthropometric measures (height, weight, and waist and hip circumference) measured by the study personnel. Finally, a blood sample was taken from each participant, aliquoted into plasma, serum, erythrocytes and white cells, and stored in liquid nitrogen at −196 °C in the study biobank. 

### 2.3. Analytical Determzination of Lead

Lead levels were measured in erythrocytes because over 95% of lead in the blood is confined in these cells [15]. Inductively coupled plasma-mass spectrometry (Thermo X7, Thermo Elemental, Winsford, UK) was used to measure lead concentration in erythrocyte samples diluted with an alkaline solution [16], with a detection limit set to 0.09 μg/L (corresponding to three times the standard deviation of the blank). Direct dilution was preferred over alternative analytical methods for its time-efficiency and because it requires fewer steps in preparation, ensuring a lower risk for contamination. Certified materials were used in all steps; analyses were conducted in accordance with the UK National External Quality Assessment Service (UK NEQAS) and the German External Quality Assessment Scheme (G-EQUAS). Since one person was selected both as a case and as a control in the EGM study, and the erythrocyte sample was not available for one more person, the present study is based on a total of 454 subjects. 

### 2.4. Reconstruction of Participants’ Residential and Occupational History

Lead tends to bioaccumulate in human tissues, and prolonged exposure can contribute to its erythrocyte levels. Therefore, we planned to contact all study participants to collect information on their residential and occupational history during the five years prior to blood taking. Participants who were still alive at the beginning of the study were contacted and invited to complete the study interview. For study participants who had already died, an attempt was made to contact a close relative (e.g., the spouse or an offspring) by asking the general practitioner of the deceased person and by mailing an invitation letter to the last known address of the deceased person. A close relative was requested to complete the study interview also on behalf of study participants who were unable to do it in person (e.g., because of dementia or stroke sequelae). Overall, the study interview was completed for 370 of 454 study subjects (81.5%), of whom 335 in person and 35 by a close relative.

The following information was collected during the study interview: complete address of each house where he/she had resided for at least six months during the study period; type of heating; whether the kitchen, bedroom, and living room were facing a street, a courtyard, a green area, etc.; whether the water used for drinking and cooking mainly came from the municipal aqueduct, a privately owned well, or was bottled water; any paid job (lasting at least six months) during the study period; for each job, details were asked regarding the type of company, workplace address, type of work and specific tasks, average number of hours worked per week; and what mean of transport was mainly used to travel to workplace. All reported occupations were classified into main groups according to the International Standard Classification of Occupations (ISCO; for consistency, the coding was made using the ISCO-68 version as this was used also at the time of EPIC cohort inception) [17]. 

### 2.5. Statistical Methods

We used the Mann-Whitney test (for binary variables) and the Kruskal-Wallis test (for non-ordered categorical variables taking three or more different values) to compare median values of erythrocyte lead levels across categories of the variables in study, and the Cuzick test to search for trend across ordered groups. 

Because of right-skewed deviations from normality, erythrocyte lead levels were logarithmically transformed (using natural logarithms, i.e. log to base e) prior to be entered as dependent variable in linear regression models. Regression coefficients were then back transformed into the original scale of erythrocyte lead levels (μg/L) and expressed as percent change to ease interpretation (see [18] for details).

Factors that were investigated as potential predictors of erythrocyte lead levels included gender and age; school level (none/primary, technical/professional, secondary school, university or higher); the year when the participant’s family first owned a car and/or a fridge (as a proxy of socioeconomic status in childhood); cigarette smoking (never, former and current), number of pack-years smoked, and (for former smokers) years since smoking cessation; parental smoking behaviour; body mass index; consumption of specific foods and food groups; physical activity levels (weekly energy expenditure, measured in metabolic equivalents, for the following activities: walking, cycling, gardening, sport, and housekeeping); and variables referring to the study participants’ residential and occupational history collected during the study interview (see above). Age at menarche, menopausal status at enrolment, parity, history of breastfeeding, and the use of oral contraceptives and hormone replacement therapy were investigated in models restricted to women. Variables included in the final models were selected via backward elimination; age, school level, and body mass index were retained in all multivariable regression models regardless of their statistical significance. Statistical interaction was tested by adding cross-term products (each at a time) for any pair of factors that were significantly associated with erythrocyte lead levels in the final models. Finally, we conducted a number of subgroup analyses (subjects who completed the study interview, never smokers, subjects included as controls in the EGM study) to test the robustness of our model.

All analyses were performed using STATA version 14 (Stata Corp, College Station, TX, USA). All analyses were two-sided, and the threshold for statistical significance of p-values was set to 0.05. 

### 2.6. Ethical Aspects and Informed Consent

The study project was approved by the committee on research ethics of the local health authority in Florence. All procedures were in accordance with the ethical standards of the institutional and/or national research committee and with the 1964 Helsinki declaration and its later amendments or comparable ethical standards. Informed consent was obtained from all individual participants included in the study at the moment of enrolment into the EPIC-Florence cohort study. 

## 3. Results

Erythrocyte lead concentrations could be determined in all of the 454 study participants. The median concentration was 86.1 μg/L (range 27.8–400.8 μg/L, inter-quartile range 65.5–111.9 μg/L), and the distribution was skewed to the right (skewness = 2.31) (Figure 1). 

Erythrocyte lead levels were higher among men than women (median 104.45 vs. 84.07 μg/L, *p* = 0.002), increased with age (69.20, 88.63, and 94.96 μg/L among those aged <45, 45–55, or >55 years, respectively, *p* for trend <0.001), were inversely associated with school level (*p* < 0.003) and were lowest among subjects with jobs typically requiring a high level of education, like professional workers or administrative and clerical staff (*p* = 0.011) (Table 1). Erythrocyte lead levels were significantly higher among former and current smokers compared to never smokers, and increased significantly with the number of pack-years among both former and current smokers; instead, they were not associated with body mass index and any index of physical activity levels. Subjects that had lived or worked in Florence for at least six consecutive months in the five years prior to blood taking had significantly higher erythrocyte lead levels compared to those who had not. Among those who declared having ever had a job (lasting at least six months) during the study period, those who used the train as means of transport to workplace had reduced erythrocyte lead levels compared to those who did not (*p* = 0.010). No other variable related with the participants’ residential and occupational history in the five years prior to blood taking was significantly associated with erythrocyte lead levels in univariate analysis. 

We reported in Table 2 the median lead erythrocyte concentration, and the corresponding IQR range, in the bottom, middle and top tertile of consumption of main foods and food groups and alcohol. Lead levels decreased with increasing consumption of vegetables (*p* = 0.021), milk and dairy products (*p* = 0.016), and fish (*p* = 0.038), and were positively correlated with alcohol intake (*p* < 0.001). In addition, there was a weak, borderline significant (*p* = 0.095) inverse association with consumption of legumes. In terms of menstrual and reproductive history and use of exogenous sex hormones, lead levels were higher among women who were already in menopause at enrolment (*p* < 0.001) and, with borderline significance, those with a late menarche (*p* = 0.071) and who had ever breastfed (*p* = 0.064) (Table 1).

In multivariable analysis (Table 3), erythrocyte lead levels were higher by 22.4% among male vs. female participants (*p* = 0.008) and increased with age at blood taking (by 31.8% for those aged >55 vs. <45 years, *p* for trend <0.001). The association with smoking habits was confirmed in multivariable analysis as well, including a dose-response association with the number of pack-years smoked among subjects who reported to be former or current smokers at enrolment. None of the association between consumption of specific foods and food groups and erythrocyte lead levels was confirmed in multivariable analysis, except for alcohol intake (<0.001). In this regard, erythrocyte lead levels were increased among wine and liquor drinkers, but not among beer drinkers (results not shown). Having ever resided in an urban environment (i.e., Florence) during the study period persisted as a predictor of erythrocyte lead levels also in multivariable analysis (−8.7%, *p* = 0.018), as well as the working condition, with professional/technical and administrative/clerical workers and retired individuals showing significantly reduced erythrocyte lead levels compared to housewives (taken as reference group). In particular a statistically significant interaction emerged between age and job condition, whereby retired people who were >55 years at enrolment into EPIC had even lower erythrocyte lead levels of those who were already retired at an earlier age (*p* for interaction 0.019). No other variable related to the participants’ residential and occupational history (including means of transport to workplace) was significantly associated with lead concentrations in multivariable analysis. 

Finally, erythrocyte lead levels were higher by 27.1% among women who were already in menopause at enrolment (*p* < 0.001) and inversely associated with the use of hormones for menopause (*p* = 0.004), while the associations with later age at menarche and breastfeeding did not persist in multivariable analysis. The adjusted R^2^ was 29.7% for the multivariable regression model fitted in the whole study sample, and 32.2% for the model fitted among women. All results were confirmed with minor changes in subgroup analyses (results not shown). 

## 4. Discussion

We studied the determinants of erythrocyte lead concentrations in a general population-based sample of 454 adults (mean age 52.2 years, 94.3% women) from Tuscany, central Italy. Some of our findings were broadly in line with previous research on the topic [19,20,21,22]. For instance, important individual predictors of erythrocyte lead levels included subjects’ age, cigarette smoking, alcohol intake, and socioeconomic status (school level and work condition) [19,20,21,22]. Previous studies found either direct [23] or inverse [24] association of erythrocyte lead levels with overweight and obesity among adults, which was not confirmed in our study. Erythrocyte lead levels were higher among subjects who had lived and/or worked in Florence (the largest city of Tuscany) at any time in the five years prior to blood taking. This was an expected finding and is most likely due to the fact that leaded gasoline was still marketed in Italy at the time of EPIC enrolment (1992–1998; it was definitively banned in 2002). In this regard, it is worth highlighting that the influence of ambient Pb on biological levels appear to have decreased over the last two decades [25], which is confirmed by recent data from several countries including Italy [19,26]. A rather unexpected finding was the complete lack of association between dietary habits and erythrocyte lead concentrations. While diet is commonly not seen as a primary source of lead in the general population, previous reports suggested that some foods or food groups may play a role, for instance fish and shellfish [26] or dairy products [27]. Instead, none of these associations was confirmed in adjusted analyses in our study, neither in the total study sample, nor in any of the subgroup analyses (including never smokers). 

Previous reports have shown somewhat heterogeneous results regarding the association of biological lead levels with gender, women’s menstrual and reproductive history, and use of exogenous sex hormones (oral contraceptives and hormones for menopause) [28,29,30]. In our study, erythrocyte lead levels were higher in men than women and, among the latter, were positively associated with a later age at menarche and an earlier menopause, and inversely associated with use of hormones for menopause. The biological mechanisms underlying this association are still poorly understood. Low iron stores modulate the expression of the divalent metal transporter 1 (DMT1) on the intestinal mucosa [31]. The DMT1 serves as transporter of other divalent metals in addition to iron, including cadmium, manganese, and others [32]. Blood levels of these metals are generally increased in low-iron states [33,34], higher among women than men [35,36], and associated with menstrual and reproductive history in women [37,38,39]. The intestinal absorption of lead is also mediated by the DMT1 [40,41], yet the association with gender and menstrual and reproductive history is in the opposite direction compared to other heavy metals. Divergent toxicokinetics of heavy metals in relation to estrogens may help explain these findings: in particular, estrogens induce deposition of lead from blood to bone, which may outweigh the effect of iron deficiency on lead absorption and blood levels [30]. It may worth noting that an effect of lead exposure on reproductive aspects was hypothesized in males as well based on the observation that lead levels in sperm negatively affects semen quality [42]. However, erythrocyte lead levels were not associated with number of children among male participants in our study (results not shown), which is in line with results from similar studies [43,44]. 

The large sample size and the wealth of information on several potential predictors of erythrocyte lead levels (pertaining to the participants’ lifestyle, dietary habits, residential and occupational history, and exposure to endogenous and exogenous estrogens) are the major strengths of our study. Some limitations also exist that need to be acknowledged. While the number of subjects included in our study exceeded that of previous publications having the same topic [20,21], our study size was not determined based on power calculations but forced to correspond to that of the EGM study, within which erythrocyte lead levels were measured. Likewise, blood samples were collected at the time of EPIC cohort inception (i.e., during the nineties), and more recent measurements were not available. As already mentioned, leaded gasoline was still in use in Italy at the time of the EPIC enrolment, and the lack of geocoded proxy measure of traffic-related air pollution (e.g., particulate matter or nitrogen dioxide) represents a possible source of misclassification of exposure, and may help explain the moderate proportion of overall variance explained by our model. Also, because of decreasing ubiquitous exposure to ambient air lead [25], the impact of lifestyle-related factors, including individual dietary habits, on erythrocyte lead levels, may be stronger today than in the past. There is evidence that early exposure to lead (e.g., those amenable to a lower socio-economic status of parents, including living in poorly maintained houses, and passive exposure to parental smoking) substantially affects blood lead levels in children [45,46,47]. As lead accumulates in the body, our lack of information on possible sources of exposure to lead in childhood may be seen as a limitation, although possibly not a major one considering that erythrocyte lead is thought to mainly reflect recent exposures [48]. Furthermore, we failed to reconstruct the residential and occupational history for some of the study participants; however, results from subgroup analyses conducted using subjects with complete data mirrored those for the whole study sample, suggesting that the impact of this limitation on results and conclusions should not be major. Finally, we were unable to assess lead exposure through eating wild game since this was not covered by the EPIC questionnaire. Hunting is widespread in rural areas of Tuscany, and evidence exists that the concentration of lead in tissue of hunted wild game (e.g., game birds, hares, and wild boars) may exceed the maximum limit in meat set by the European Commission [49,50], and that regular consumption of wild game may be associated with higher lead blood levels [51], so that this topic would deserve to be investigated in future studies.

## 5. Conclusions

In conclusion, we found that a substantial proportion of inter-individual variation in erythrocyte lead levels in adults could be accounted for by demographic factors (age and gender), smoking habits, socioeconomic status, alcohol intake, place of residence, and (among women) menstrual and reproductive factors. Environmental exposure to lead has decreased in recent decades following the ban of leaded gasoline in the 90s and the prohibition made in the 80s to use lead paints for new houses and in the restoration and repainting of old houses. As a consequence, non-environmental sources of exposure to lead are likely to play an increasingly important role in determining biological lead levels in the general population. As already mentioned, convincing evidence exists that exposure to low- or even very low-level lead may have important adverse effects on health (NTP 2012), especially due to its ability to impair kidney function and cause hypertension [4,52]. Therefore, studying the main determinants of lead body burden in the general population remains an important research priority in order to implement public health measures aimed at effectively reducing lead exposure. 

## Figures and Tables

**Figure 1 ijerph-16-00425-f001:**
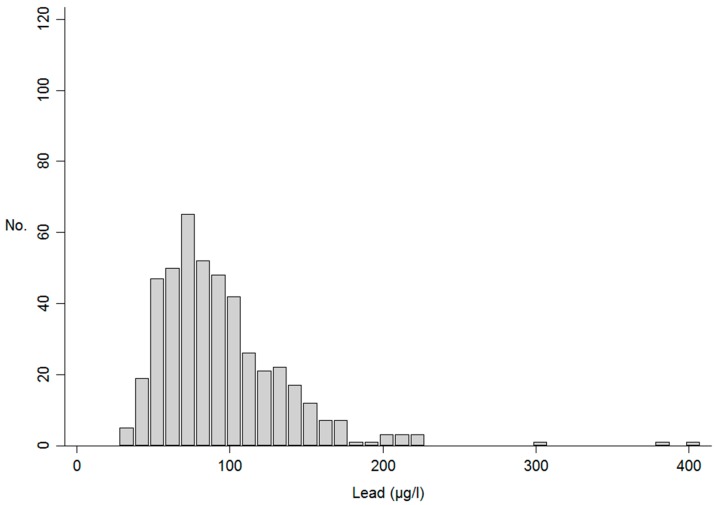
Erythrocyte lead levels (μg/L) among the 454 subjects included in the study.

**Table 1 ijerph-16-00425-t001:** Erythrocyte lead levels (μg/L) according to selected participants’ characteristics.

Participants’ Characteristics	No.	%	Lead (μg/L)	
			Median (IQR)	*p*-Value ^(a)^
**Total**	454	100.0%	86.07 (65.53–111.93)	-
**Gender**				
Male	26	5.7%	104.45 (85.09–137.54)	
Female	428	94.3%	84.07 (64.05–110.57)	0.002
**Age**				
<45 years	107	23.6%	69.20 (55.61–82.46)	
45–55 years	164	36.1%	88.63 (66.70–111.38)	
>55 years	183	40.3%	94.96 (74.54–123.18)	<0.001
**Education level**				
None/primary school	129	28.5%	94.42 (71.04–116.50)	
Technical/professional school	83	18.3%	87.30 (62.24–110.73)	
Secondary school	174	38.4%	80.60 (62.23–113.58)	
University or higher degree	67	14.8%	75.05 (61.89–103.94)	0.003
**Work condition**				
Housewife	114	25.1%	92.80 (65.10–119.70)	
Retired	75	16.5%	92.37 (73.09–118.39)	
Unemployed	7	1.5%	100.55 (68.48–216.21)	
Professional, technical and related	78	17.2%	75.19 (63.04–105.54)	
Administrative, manager, clerical	101	22.2%	75.49 (59.19–95.11)	
Sales workers	29	6.4%	99.23 (75.45–122.80)	
Service workers	17	3.7%	80.34 (69.91–133.16)	
Production, transport, labourers	33	7.3%	90.42 (73.63–134.20)	0.011
**Smoking habits**				
Never smoker	209	46.0%	80.34 (62.23–104.73)	
Former smoker	118	26.0%	87.89 (68.87–115.63)	0.025
Current smoker	127	28.0%	92.88 (70.87–128.58)	0.004
**Pack years**				
Former smoker, 1st tertile	40	34.8%	81.72 (58.35–96.30)	
2nd tertile	37	32.2%	82.96 (68.34–112.48)	
3rd tertile	38	33.0%	108.08 (78.80–139.11)	0.003
Current smoker, 1st tertile	42	33.9%	73.17 (62.70–93.26)	
2nd tertile	41	33.1%	100.35 (70.87–134.33)	
3rd tertile	41	33.1%	101.15 (82.72–139.58)	0.001
**Body mass index**				
<25	248	54.9%	80.43 (62.06–105.39)	
25–30	161	35.6%	93.26 (70.95–124.96)	
>30	43	9.5%	80.34 (60.36–102.21)	0.092
**Living in Florence (anytime during the study period)**				
Yes	329	72.5%	91.25 (68.34–115.63)	
No	125	27.5%	73.96 (59.47–95.16)	0.001
**Working in Florence (anytime during the study period)**				
Yes	170	76.6%	80.50 (63.47–107.65)	
No	52	23.4%	70.99 (55.89–89.52)	0.024
**Driving to workplace**				
No	102	43.6%	82.43 (66.60–105.54)	
Yes	132	56.4%	73.97 (61.01–100.06)	0.334
**By train to workplace**				
No	221	94.4%	80.40 (63.04–103.46)	
Yes	13	5.6%	56.68 (48.21–78.03)	0.010
**Walking to workplace**				
No	175	74.8%	75.49 (59.24–103.90)	
Yes	59	25.2%	81.45 (72.01–101.49)	0.176
**Women (n = 428)** **Age at menarche**				
≤12 years	223	52.2%	80.25 (60.77–110.71)	
≥13 years	204	47.8%	89.47 (68.49–109.69)	0.071
**Menopausal status**				
Pre- or peri-menopausal	196	45.8%	71.07 (55.76–91.14)	
Post-menopausal	232	54.2%	97.66 (77.23–126.49)	<0.001
**Full-term pregnancies**				
None	20	4.7%	79.14 (59.39–102.30)	
1	138	32.2%	80.92 (62.29–109.18)	
2	179	41.8%	82.47 (63.46–110.73)	
≥3	91	21.3%	93.03 (68.10–113.25)	0.132
**Breastfeeding**				
Never	334	81.9%	82.59 (63.04–107.65)	
Ever	74	18.1%	93.10 (72.15–113.58)	0.064
**Oral contraceptives**				
Never	230	53.7%	86.05 (68.26–106.64)	
Ever	198	46.3%	80.60 (61.89–112.00)	0.355
**Hormones for menopause**				
Never	175	75.4%	99.64 (78.01–130.59)	
Ever	57	24.6%	93.03 (76.41–112.13)	0.128

^(a)^ Medians were compared using the Mann-Whitney test (binary variable), the Kruskal-Wallis test (non-ordered categorical variables taking three or more different values), or the Cuzick test (trend across ordered groups); ^(b)^ For six consecutive months or longer.

**Table 2 ijerph-16-00425-t002:** Erythrocyte lead levels (μg/L) according to the consumption of selected foods and food groups.

Selected Foods and Food Groups	Lead (μg/L), Median Values (IQR)			
	1st Tertile	2nd Tertile	3rd Tertile	*p*-Value ^(a)^
Vegetables	91.12 (72.44–118.46)	82.72 (62.23–108.33)	80.60 (62.24–109.44)	0.021
Olive oil	90.45 (69.46–113.46)	81.45 (64.65–110.39)	84.77 (63.55–110.73)	0.306
Fruit	90.10 (67.06–118.14)	83.68 (65.42–108.76)	81.35 (65.02–110.44)	0.172
Legumes	85.55 (66.80–120.21)	89.79 (69.73–112.35)	81.45 (62.23-106.38)	0.095
Pasta and rice	88.63 (69.52–106.71)	87.60 (63.55–125.36)	79.92 (62.70–106.73)	0.173
Mushrooms	86.69 (69.20–113.25)	87.54 (66.80–116.36)	82.24 (60.36–101.35)	0.204
Milk and dairy products	91.28 (71.07–123.91)	81.32 (62.64–107.65)	83.04 (63.83–107.05)	0.016
White meat	89.68 (68.78–113.27)	82.85 (66.23–109.52)	86.28 (61.88–111.66)	0.247
Red meat	78.46 (62.06–108.89)	87.91 (67.71–112.00)	88.84 (66.80–116.50)	0.168
Processed meat	88.61 (65.97–110.67)	85.09 (66.60–112.48)	83.23 (63.55–111.66)	0.652
Fish	88.84 (70.63–113.58)	88.78 (68.41–111.19)	78.13 (60.77–110.71)	0.038
Crustaceans and molluscs	84.95 (68.87–112.35)	86.06 (65.23–109.27)	86.09 (61.23–113.25)	0.489
Alcohol	76.28 (58.71–101.49)	80.40 (63.46–103.90)	100.35 (78.13–135.85)	<0.001
Energy intake	88.61 (68.67–110.67)	86.14 (68.01–121.46)	81.97 (61.67–110.73)	0.271

^(a)^*p*-values were calculated using the non-parametric Cuzick test for trend of medians across ordered groups.

**Table 3 ijerph-16-00425-t003:** Association between selected participants’ characteristics and erythrocyte lead levels.

Participants’ Characteristics	Erythrocyte Lead Levels (μg/L)		
	Percent Change	95% CI	*p*-Value (for Trend)
**All study sample (*n* = 454)**			
Gender			
Female	ref		
Male	22.4%	(5.4%, 42.0%)	0.008
**Age**			
<45 years	ref		
45–55 years	20.5%	(10.3%, 31.6%)	
>55 years	31.8%	(19.1%, 45.8%)	<0.001
**Smoking habits**			
Never smoker	ref		
Former smoker, 1st tertile PY	2.7%	(−8.8%, 15.6%)	
2nd tertile PY	7.3%	(−5.0%, 21.1%)	
3rd tertile PY	22.2%	(7.8%, 38.5%)	0.006
Current smoker, 1st tertile PY	−1.9%	(−12.5%, 10.0%)	
2nd tertile PY	21.8%	(8.5%, 36.6%)	
3rd tertile PY	23.0%	(9.5%, 38.1%)	<0.001
**Alcohol intake**			
1st tertile	ref		
2nd tertile	12.2%	(3.7%, 21.4%)	
3rd tertile	40.8%	(29.8%, 52.7%)	<0.001
**Living in Florence during the study period**			
Ever	ref		
Never	−8.7%	(−15.3%, −1.5%)	0.018
**Work condition**			
Housewife	ref		
Retired	−10.4%	(−19.2%, −0.6%)	0.039
Unemployed	22.1%	(−6.0%, 58.6%)	0.135
Professional, technical and related	−13.0%	(−21.4%, −3.7%)	0.007
Administrative, manager, clerical	−11.9%	(−19.9%, −3.1%)	0.009
Sales workers	1.7%	(−12.1%, 17.6%)	0.819
Service workers	6.5%	(−10.6%, 26.9%)	0.480
Production, transport, labourers	−1.4%	(−14.3%, 13.4%)	0.846
**Women *(n* = 428)**			
**Menopausal status**			
Pre- or peri-menopausal	ref		
Post-menopausal	27.1%	(16.2%, 40.5%)	<0.001
**Use of hormones for menopause**			
Never	ref		
Ever	−13.9%	(−21.3%, −4.9%)	0.004

Multiple linear regression model with natural logarithm-transformed lead levels as dependent variable; further adjusted by school level, energy intake, and body mass index. Regression coefficients were back transformed into the original scale (μg/L) and expressed as percent change (see [ref] for details). CI: confidence intervals.

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
