# Peer review of "Determinants of Erythrocyte Lead Levels in 454 Adults in Florence, Italy"

_ijerph, 2019, doi:10.3390/ijerph16030425_

Round 1
Reviewer 1 Report
Thanks for the opportunity to review this excellent article. My comments can be found below:
Introduction: You should also speak to leads effects on the hepatobiliary system as this is relevant to your study:
Obeng-Gyasi, E., Armijos, R.X., Weigel, M.M., Filippelli, G. and Sayegh, M.A., 2018. Hepatobiliary-Related Outcomes in US Adults Exposed to Lead. Environments, 5(4), p.46.
Dioka, C.E., Orisakwe, O.E., Adeniyi, F.A.A. and Meludu, S.C., 2004. Liver and renal function tests in artisans occupationally exposed to lead in mechanic village in Nnewi, Nigeria. International journal of environmental research and public health, 1(1), pp.21-25.
In addition speak to to leads effects during the life course:
Reuben, A., Caspi, A., Belsky, D.W., Broadbent, J., Harrington, H., Sugden, K., Houts, R.M., Ramrakha, S., Poulton, R. and Moffitt, T.E., 2017. Association of childhood blood lead levels with cognitive function and socioeconomic status at age 38 years and with IQ change and socioeconomic mobility between childhood and adulthood. Jama, 317(12), pp.1244-1251.
Obeng-Gyasi, E., 2018. Lead Exposure and Oxidative Stress-A Life Course Approach in US Adults. Toxics, 6(3).
Methods:
Did you do a power analysis to see if you had an adequate sample size for the effects you are trying to detect? Please state that in the manuscript.
Results:
Well done
Discussion:
Well done.
Author Response
1. Introduction: You should also speak to leads effects on the hepatobiliary system as this is relevant to your study:
Obeng-Gyasi, E., Armijos, R.X., Weigel, M.M., Filippelli, G. and Sayegh, M.A., 2018. Hepatobiliary-Related Outcomes inUSAdults Exposed to Lead. Environments, 5(4), p.46.
Dioka, C.E., Orisakwe, O.E., Adeniyi, F.A.A. andMeludu,S.C., 2004. Liver and renal function tests in artisans occupationally exposed to lead in mechanic village inNnewi,Nigeria. International journal of environmental research and public health, 1(1), pp.21-25.
In addition speak to lead effects during the life course:
Reuben, A., Caspi, A., Belsky, D.W., Broadbent, J., Harrington, H., Sugden, K., Houts, R.M., Ramrakha, S., Poulton, R. and Moffitt, T.E., 2017. Association of childhood blood lead levels with cognitive function and socioeconomic status at age 38 years and with IQ change and socioeconomic mobility between childhood and adulthood. Jama, 317(12), pp.1244-1251.
Obeng-Gyasi, E., 2018. Lead Exposure and Oxidative Stress-A Life Course Approach in US Adults. Toxics, 6(3).
Response: Thank you for this suggestion. We added some text in the Introduction, and the corresponding references, as suggested by the Reviewer.
2. Methods: Did you do a power analysis to see if you had an adequate sample size for the effects you are trying to detect? Please state that in the manuscript.
Response: Thank you for this comment. The present study takes advantage of data relating to the EPIC-Florence centre that were collected at enrolment into the EPIC cohort and during Envirogenomarkers (EGM), a case-control study nested within EPIC. In particular, the EGM study aimed to study the association between a number of pollutants (including lead) measured in blood samples collected at enrolment within the EPIC study (i.e. in 1992-98), and the risk of subsequent breast cancer or non-Hodgkin lymphoma. The present study represents a secondary analysis of the data collected within the EGM study, whereby erythrocyte lead levels were considered as the outcome of interest, and the study objective was to identify their main predictors. Since lead levels were not measured in blood of all EPIC subjects (which would not be feasible because of the high costs), but only among those included in the EGM study, the number of subjects included was pre-defined (i.e. not based on power calculations) and forced to correspond to the number of subjects from the EPIC-Florence centre that were included in the EGM study. Although our study size exceeded that of previous publications with a similar topic, the above may be seen as a limitation, which we decided to acknowledge in the Discussion (limitations section).
Results: Well done.
Discussion: Well done.

Reviewer 2 Report
This deals with an interesting question and has some interesting findings However, I have some problems with it. I don’t understand what erythrocyte lead levels (ELL are. I am very familiar with Blood Lead Levels (BLL) but not of ELL. What is their validity? What is their relation to BLL?
The ELL’s reported are much higher that the median BLL at that time in the US ( I am aware that these reports are mcg/L vs mcg/dL which is reported in the US.) Are those levels really much higher or is this a result of ELL vs BLL?
The data is 20 -25 years old. In the US, BLL went down substantially between 1998 and 2005. I would be interested in seeing some more recent date
There are many important US publications that predict BLL on the basis of social and community factors that are not cited. In particular, look at the writings of Lanphear, of Krieger and of Kaplowitz.
How did the study get from over 13,000 eligible participant to just 454 actual ones?
I agree with the authors treating ELL as a continuous variable and log transforming it. I think the multiple regression results are important- but question the value of the bivariate results
While many of the epidemiological finding are of interest, it would be better if the ms distinguished between those that are new and those that confirm previous research. It would also be better if the authors did more explaining of their findings and trying to give tem some unifying conceptualizing, For example much of the literature in the US see low income people living in poorly maintained houses, built before 1940 (when paint had a high concentration of lead) as creating the greatest risks.
Author Response
Reviewer 2
This deals with an interesting question and has some interesting findings. However, I have some problems with it.
1. I don’t understand what erythrocyte lead levels (ELL) are. I am very familiar with Blood Lead Levels (BLL) but not of ELL. What is their validity? What is their relation to BLL?
Response: Erythrocyte lead levels (ELL) were preferred over whole blood lead levels (BLL) because more than 95% of lead in the blood is confined within erythrocytes (https://www.ncbi.nlm.nih.gov/pubmed/17225467), while less than 1% is found in plasma. We mentioned this in the Methods (Analytical determination of lead, first sentence), to explain why ELL was measured and used for the analyses, and added the above reference. Please also note that some evidence exists that ELL may be a “more effective biomarker to interpret the hematotoxicity of lead” (https://www.ncbi.nlm.nih.gov/pubmed/25588596). In addition, other studies have used ELL (instead over BLL) before (for instance here: https://www.ncbi.nlm.nih.gov/pubmed/29699886).
2. The ELL’s reported are much higher that the median BLL at that time in theUS(I am aware that these reports are mcg/L vs. mcg/dL which is reported in theUS). Are those levels really much higher or is this a result of ELL vs. BLL?
Response: We confirm that the differences are due to the fact that lead was measured in erythrocytes instead of whole blood (considering that erythrocytes constitute only a variable percentage of blood volume). This may be seen, for instance, in Table 1 of this paper: https://www.ncbi.nlm.nih.gov/pubmed/25588596. Therefore, although ELL and BLL are (of course) correlated, they cannot be compared directly.
3. The data is 20-25 years old. In theUS, BLL went down substantially between 1998 and 2005. I would be interested in seeing some more recent data.
Response: Thank you for this comment. The present study takes advantage of data relating to the EPIC-Florence centre that were collected (1) at enrolment into the EPIC cohort, and (2) during Envirogenomarkers (EGM), a case-control study nested within the EPIC cohort. The EGM study aimed to study the association between a number of pollutants (including lead) measured in blood samples collected at enrolment within the EPIC study (i.e. in 1992-98), and the risk of subsequent breast cancer or non-Hodgkin lymphoma. The present study represents a secondary analysis of the data collected within the EGM study, whereby erythrocyte lead levels were considered as the outcome of interest, and the study objective was to identify their main predictors. The above justifies the fact that (1) erythrocyte lead levels were measured in blood samples collected in the 90s (i.e. at the time of the EPIC cohort inception), and no later measurements were available; and (2) the number of subjects included was pre-defined and corresponded to the number of subjects from the EPIC-Florence centre that were included in the EGM study. We added some text in the Discussion (limitations section) to acknowledge the above.
4. There are many importantUSpublications that predict BLL on the basis of social and community factors that are not cited. In particular, look at the writings of Lanphear, of Krieger and of Kaplowitz.
Response: Thank you for your insightful comment. We agree with the Reviewer that a vast scientific literature shows how blood lead levels in children are affected by factors linked to the parents’ socioeconomic status (e.g. the parents’ school level, living in an older and poorly maintained house, and being passively exposed to the parents’ smoking). Since lead accumulates in the body, the effect of early exposure may indeed extend to adulthood (although blood lead levels are thought to reflect recent exposures, see for instance here: https://www.ncbi.nlm.nih.gov/pmc/articles/PMC2533151/, so perhaps this is not a major limitation in our case). Unfortunately, the only variables of the like that were collected within the EPIC study were the parent’s smoking habits and the year in which a fridge and/or a car were first owned by the participant’s family. None of these variables were statistically associated with erythrocyte lead levels in our sample, nor increased in a significant way the variance explained by the model. However, we added some text in the Discussion to mention some of the articles suggested by the reviewer and to acknowledge our relative lack of information on potential exposures occurring early in one’s life as a potential limitation of our study. Text was also added to the Methods to mention the new variables that were tested.
5. How did the study get from over 13,000 eligible participants to just 454 actual ones?
Response: Please refer to our response to comment #3 above for a detailed explanation. Briefly, the EPIC-Florence cohort encompasses over 13,000 subjects, but only 454 were included in the Envirogenomarkers (EGM) case-control study nested in the cohort. Lead levels were not measured at EPIC cohort inception, but only during the EGM study (i.e. only among subjects included in the EGM study). This explains why the number of subjects included in the present study is 454 instead of 13,000. We added some text in the Discussion (limitations section) to acknowledge the above.
6. I agree with the authors treating ELL as a continuous variable and log transforming it. I think the multiple regression results are important - but question the value of the bivariate results.
Response: Based on this comment, and on comment #4 by Reviewer #3 (see below), we decided to convert the regression coefficients (which are expressed as additive change to the log of erythrocyte lead levels) back to their “natural” scale. This implies that the regression coefficients are now expressed in percent change (see for instance here for details: https://www.cscu.cornell.edu/news/statnews/stnews83.pdf). This rendered the outputs of the regression models more comparable with the bivariate results. As a result, many changes had to be made in the text (in particular in the Results section) to adapt it to the changed form of presenting the results.
7. While many of the epidemiological finding are of interest, it would be better if the authors distinguished between those that are new and those that confirm previous research.
Response: Thank you for this comment. We made changes to the first and second paragraph of the Discussion to make it clear what results confirm previous research and what are new and/or unexpected. We also added some text (and the corresponding references) to compare our negative results on the association between lead levels and anthropometric measures with previous literature on the topic.
8. It would also be better if the authors did more explaining of their findings and trying to give them some unifying conceptualizing. For example, much of the literature in theUSsee low income people living in poorly maintained houses, built before 1940 (when paint had a high concentration of lead) as creating the greatest risks.
Response: Thank you for this suggestion. As mentioned above, we did not have information on the year each participant’ house was built, nor whether it had been restored at some time (including during the study period). Likewise, we were unfortunately unable to investigate the role of exposures back in the childhood on current (i.e. in adulthood) lead levels. However, considering that the environmental exposure to lead is very likely to have declined over the last couple of decades in Italy (because of the ban of lead gasoline in the 90s, and because lead-free paints have been used since the 80s – i.e. for 30-40 years - for new houses and in the renovation and repainting of old houses), it seems reasonable to assume that the relative importance of lifestyle-related exposures to lead may have increased over time and be quite important in our days (i.e. an increasing share of lead that enters our body, does it through lifestyle-related sources of exposure, we believe). This is, in your opinion, the overall picture that we can draw from our results, and we modified the concluding paragraph to state it more clearly.

Reviewer 3 Report
Please see the attached file.

Author Response
This study investigated the determinants of individual lead levels in a general population based sample of 454 adults enrolled in the Florence cohort of the European Prospective Investigation into Cancer and Nutrition (EPIC) study, and found that male gender, older age, cigarette smoking and number of pack-years, alcohol intake, and residing in urban areas were positively associated with higher erythrocyte lead levels, while performing professional/managerial or administrative work or being retired was inversely associated with lead levels. These results provide information for avoidable risk factors contributing to the lead body burden among adults, which could therefore be lowered through targeted public health measures. The manuscript is well prepared and the results are informative. I have the following comments to improve the manuscript based on its present form.
1. The study investigated the associations between erythrocyte lead levels and multiple factors, and it would be necessary to account for the issue of multiple testing when interpret the results. Some of the marginal significant results could be due to chance finding.
Response: Thank you for this comment. The Authors have thoroughly discussed this point and, after an intense debate, the final decision was not to use any adjustment for multiple comparisons. There are some reasons that led us to take this decision. The most important one is that our study cannot be seen as a “fishing expedition”, i.e. a search for novel statistical associations that may help generate new hypotheses. On the contrary, all of the factors that were investigated (gender, age, proxies of socioeconomic status, smoking, anthropometric measures, consumption of specific foods and food groups, occupational and residential history, and hormone-related variables among women) were found to be associated with biological blood levels in past studies, and our objective was to verify which among those potential sources of exposure played a significant role in our setting as well. Since all hypotheses that were tested had an established (i.e. literature-based) rationale, we believe no adjustment is needed, as we might risk missing some important associations. Technically, by reducing type I error, we would have increased type II error, but considering that we were interested in evaluating the percent of variance explained by the above variables (instead of “discovering” new associations), that was something we definitely wanted to avoid, as it would have led to possibly incorrect conclusions. The above considerations were also based on recommendations made by several authors (epidemiologists and statisticians) in past years, see for instance here: https://www.ncbi.nlm.nih.gov/pubmed/2081237; and here: https://www.ncbi.nlm.nih.gov/pubmed/24697967. We hope this decision of the Authors is acceptable for the Reviewer.
2. Need to specify the value of logarithmic base for logarithmical transformation. Currently the base value is unknown and the regression coefficients shown in Table 3 are difficult to understand.
Response: We thank the Reviewer for spotting this inaccuracy. We specified in the text (statistical methods) and in the footnote of Table 3 that we used natural logarithms (ln, i.e. log to base e).
3. Suggest add lead levels of 2nd tertiles in Table 2 so the readers are easy to get the trends in lead levels over the tertiles.
Response: We accepted this comment and modified Table 2 as suggested. We also slightly modified the text and the table footnote for consistency.
4. The effect estimates shown in Table 3 could be more informative if changed to percent changes rather than regression coefficients. Consider antilogarithmic conversion of the regression coefficients.
Response: Many thanks for this suggestion. Based on this comment, and on comment #6 by Reviewer #2 (see above), we decided to convert the regression coefficients (which are expressed in additive change to the log of erythrocyte lead levels) back to the natural scale. As noted by this reviewer, this implies that the regression coefficients are now expressed in percent change (see for instance here for details: https://www.cscu.cornell.edu/news/statnews/stnews83.pdf). As a result, changes had to be made in the text (in particular in the Results section) to adapt it to the changed form of presenting the results.
5. It would be interesting to investigate any potential interactions between the significant factors associated with lead levels. Could add a cross-product term for any two factors each at a time in the final model. This analysis may provide important information regarding the mutual impact of co-existing factors.
Response: We accepted this suggestion and added cross-product terms (each at a time) for any two significant factors in the model, and checked whether the cross-product terms were statistically significant and the variance explained by the model increased. The only statistically significant interaction was found between age and job condition, showing that erythrocyte lead levels were even lower among retired people who were older at enrolment into the EPIC study. We modified the text (Statistical methods and Results) as necessary to report this result.

Round 2
Reviewer 1 Report
The article is ready for publication.Reviewer 2 Report
I have seen the revisions and the explanation of them and they met my concersn
Reviewer 3 Report
NA